# Levels and Health Risk Assessment of Polycyclic Aromatic Hydrocarbons in Vegetable Oils and Frying Oils by Using the Margin of Exposure (MOE) and the Incremental Lifetime Cancer Risk (ILCR) Approach in China

**DOI:** 10.3390/foods12040811

**Published:** 2023-02-14

**Authors:** Qing Liu, Pinggu Wu, Pingping Zhou, Pengjie Luo

**Affiliations:** 1Key Laboratory of Food Safety Risk Assessment, Ministry of Health, China National Center for Food Safety Risk Assessment, Beijing 100021, China; 2Zhe Jiang Provincial Center for Disease Control and Prevention, Hangzhou 310051, China

**Keywords:** frying oil, health risk, incremental lifetime cancer risk (ILCR), margin of exposure (MOE)

## Abstract

A total of 139 vegetable oils and 48 frying oils produced in China were tested for the levels of 15 Environmental Protection Agency-regulated polycyclic aromatic hydrocarbons (PAHs). The analysis was completed by high-performance liquid chromatography-fluorescence detection (HPLC-FLD). The limit of detection and limit of quantitation were ranged between 0.2–0.3 and 0.6–1 μg/kg, respectively. The average recovery was 58.6–90.6%. The highest mean of total PAHs was found in peanut oil (3.31 μg/kg), while the lowest content was found in olive oil (0.39 μg/kg). In brief, 32.4% of vegetable oils exceeded the European Union maximum levels in China. The detected level of total PAHs in vegetable oils was lower than the frying oils. The mean dietary exposure to PAH15 ranged from 0.197 to 2.051 ng BaPeq/kg bw/day. The margin of exposure values was greater than 10,000, and the cumulative probabilities of the incremental lifetime cancer risk of different age groups were less than the priority risk level (10^−4^). Therefore, there was no potential health concern for specific populations.

## 1. Introduction

Polycyclic aromatic hydrocarbons (PAHs) are a family of organic or synthetic contaminants with chemically stable lipids that are comprised of two or more fused aromatic rings of carbon and hydrogen, and they do not contain heteroatoms. They widely exist in the environment (e.g., land and water) and food (e.g., vegetables) [1]. PAHs are formed by organic pyrolysis and incomplete combustion [2,3,4]; that is, natural sources, such as biosynthetic reactions, forest fires, and volcanic eruptions, form PAHs, as well as anthropogenic sources, such as the incomplete combustion of fossil fuels (oil and coal), organic matter (wood and tobacco), and automobile exhaust emissions [5]. PAHs in the environment exist at low levels. However, in recent years, with the continuous development and growth of economies, environmental pollution has increased, and environmental issues have become serious, and deserve urgent attention to the increased levels of PAHs [6,7,8,9]. In China, the total annual emission of PAHs was approximately 120,000 tons in 2012, and many human specimens showed elevated levels of PAHs [10,11]. PAHs in the atmosphere, soil, and water can be absorbed by plants, and then transferred to animals through the food chain. Therefore, raw foods contain PAHs. Processing processes (such as baking, frying, and smoking) at high temperatures are also major sources generating PAHs in food [12]. For non-smokers, PAHs enter the body through food, and significantly, vegetable oils and other fats are the primary contributors [7,13,14].

PAHs are endocrine disruptors, and they possess carcinogenic, teratogenic, and mutagenic properties [15,16,17]. They also cause harmful effects on cardiovascular and respiratory systems [8,18]. Liu and colleagues reported that exposure to PAHs can cause DNA oxidative damage in Chinese coke-oven plant workers [19]. As an occupational group with long-term PAH exposure, the incidence of lung cancer in coke-oven workers is significantly higher than that in the general population. Exposure to PAHs during pregnancy is a risk factor for abnormal neurobehavioral development [8]. Furthermore, it has been demonstrated that childhood asthma is associated with exposure to PAHs [20]. Due to the adverse effects of PAHs, many countries (agencies) around the world have built or proposed regulatory limits for PAHs in various foods. The European Commission recognized the maximum permitted levels (EU limits) of PAHs in oils and fats as 2 μg/kg for BaP and 10 μg/kg for PAH_4_ (sum of benzo[*a*]anthracene (BaA), chrysene (CHr), benzo[*b*]fluoranthene (BbF), and Benzo[*a*]pyrene (BaP)) (EC/835/2011 2011) [21]; however, China established the maximum permitted level of BaP in oils as 10 μg/kg (GB2762-2012) [22]. However, no maximum permitted level of PAH_4_ has been proposed by Chinese regulations (CN limit), and the EU limits were used for reference in this study.

Many analytical techniques were performed to detect and measure the PAHs in different food products, using GC–MS/FID (gas chromatography equipped with mass spectrometry or flame ionization detector), and HPLC-UV/FLD (high-performance liquid chromatography equipped with ultraviolet or fluorescence detectors) [11,18]. Fluorescence detection is the most widely extended technique for the detection of PAHs after their separation by LC. The extraction of PAHs from food is crucial in the multi-stage sample preparation procedure. Gel permeation chromatography combines separation and purification, and the method is simple, rapid, easy to operate, automatic, has a high recovery rate, good repeatability, and good stability. It is widely used in the separation and purification of PAHs in vegetable oils.

The contamination of vegetable oils containing PAHs is a worldwide problem, and several studies have measured the levels of PAHs in oils and estimated the dietary intake of PAHs in the general population [23,24]. These studies concluded that the adverse effects of vegetable oils on human health should be of great concern. Furthermore, reliable data on the contamination and intake of vegetable oils are needed to assess exposure risk and to revise national food standards [25]. In China, several studies have reported the contamination of vegetable oils with PAHs only in certain regions [26,27,28,29]. Another study revealed that frying or grilling increases PAH levels, and PAHs with a mutagenic ability are formed when oils are repeatedly heated to high temperatures such as at restaurants [14,24,30]. Fried foods are eaten by many individuals because of their crispy taste and attractive aroma, and elevated levels of PAHs are present not only in the fried foods but also in the frying oils. Frying oil is the vegetable oil after frying food. People usually use cheap vegetable oils to fry food, such as soybean oil or blended oil. After repeated high-temperature frying, it will definitely affect the content of PAHs. It would be useful to study the contamination of PAHs in frying oils used in restaurants in China. This is the first time the frying oils used in Chinese restaurants have been analyzed. Furthermore, the analysis focused on five regions of oil production to ensure that the whole country was covered.

In addition, the margin of exposure (MOE) and the incremental lifetime cancer risk (ILCR) were used to perform the risk assessment in vegetable oils for PAHs in preschoolers, school-age youths, and adults by the Monte Carlo simulation (MCS) method. These two health risk assessments were calculated based on the BaP toxic equivalency quotient (TEQBaP), including PAH15 in vegetable oils available in the Chinese market.

## 2. Materials and Methods

### 2.1. Sampling

The data on PAHs were acquired from the National Food Contaminant Information System in 2017. A total of 139 samples, including vegetable oils (91) and frying oils (48), were collected and analyzed for PAHs. In this study, the samples were collected from local markets and supermarkets in Hebei, Hunan, Jiangsu, Shandong, and Zhejiang provinces.

### 2.2. Chemicals and Reagents

In brief, a mixed standard solution of 200 μg/mL PAH15 (benz[*a*]anthracene(BaA), chrysene(Chr), benzo[*b*]fluoranthene(BbF), benzo[*a*]pyrene (Bap), benzo[*k*]fluoranthene(BkF), dibenz[ah]anthracene(DahA), benzo[*ghi*]perylene(BghiP), indeno [123cd]pyrene(Icdp), pyrene(Pyr), fluoranthene(Flt), anthracene(Ant), phenanthrene(Phe), fluorine(Flo), acenaphthene(Ace), naphthalene (Nap)) was used. EPA 610 110121 (Environmental Protection Agency) was purchased from Dr. Ehrenstorfer GmbH (Augsburg, Germany). HPLC-grade ethyl acetate, acetonitrile, and cyclohexane were purchased from J.T. Baker (Leicester, England). Olive oil (PAHs quality control material, no. T0665QC) was purchased from Fapas (Sand Hutton, York, UK). Working solutions of the 15 PAHs were prepared in acetonitrile (100 ng/mL). All working standard solutions were freshly prepared weekly and stored at −4 °C.

### 2.3. Sample Analysis

A validated HPLC-FLD method with some modifications was used for the determination of PAHs in vegetable oils [31].

The vegetable oil samples (2 g) were put into 15 mL polypropylene centrifuge tubes, and then 10 mL of ethylene acetate:cyclohexane (1:1, *v/v*) was added. The samples were vortexed for 1 min and sonicated for 15 min. After centrifugation (4500 r/min for 3 min), the supernatant was collected and purified. The upper phase was placed into a 5 mL vial and the sample was injected in the gel permeation chromatography (GPC) system with a purification column (200 mm × 25 mm, styrene divinylbenzene copolymer gels, Bio-Beads S-X3). Ethylene acetate: cyclohexane (1:1, *v/v*) was used as the mobile phase at a column flow rate of 5 mL/min. The eluate containing the target PAH was collected from 19 to 50 min, evaporated, and concentrated into a small volume using a vacuum rotary evaporator (Rotavapor R-300, Buchi, Flawil, Switzerland). After that, it was transferred and dried to near dryness using a pressure blowing concentrator and re-dissolved in 0.5 mL of acetonitrile. Finally, the extract was transferred into an auto-sampler vial for determination by HPLC-FLD.

### 2.4. Apparatus Conditions

The analytes were separated on a C18 column (250 mm × 4.6 mm, 5 μm) using acetonitrile and water as the mobile phase for gradient elution. The detection was performed using fluorescence detection. The mobile phase flow rate was 1.5 mL/min. The column temperature was 30 °C. The gradient was as follows: 0–5 min, 50% acetonitrile; 5–20 min, 100% acetonitrile; 20–26 min, 100% acetonitrile; 26–27 min, 50% acetonitrile; and 27–30 min, 50% acetonitrile. The detection wavelengths of the detectors are listed in Table 1. The external standard was adopted for quantification.

The HPLC system was calibrated using an external standard. The concentration ranges of the calibration standards were 0, 0.5, 1, 5, 10, and 20 ng/mL. There was good agreement between the standard and sample chromatograms on any given day.

### 2.5. Method Validation

The laboratories were quality-control certified in the detection procedure by the China National Center for Food Safety Risk Assessment (CFSA). This ensured the accuracy and comparability of the monitoring data between different laboratories. We used the isotope-labeled internal standards for quantification. LOD and LOQ were determined to be a signal-to-noise ratio of 3 and 10, respectively. The LOD and LOQ of PAHs were 0.2–0.3 and 0.6–1 μg/kg, respectively. Three different concentrations (3.0, 5.0, and 10.0 μg/kg) were spiked into blank vegetable oil samples to evaluate sample recovery. The average recovery was 58.6–90.6%, in line with the standards set out in the European Union (EU) Regulation 836/2011 (recovery between 50% and 120%). The relative standard deviation was <15% for all analyses. One quality control sample of cocoa butter (FAPAS0671) was used from the Food Analysis Performance Assessment Scheme (The Food and Environment Research Agency, Sand Hutton, York, UK). The certified values for BaA, Chr, BbF, BkF, BaP, DBahA, BghiP, and IcdP were 1.39–3.56, 2.20–5.66, 1.52–3.90, 0.73–1.87, 2.42–6.23, 0.67–1.72, 1.08–2.76, and 0.84–2.16, respectively, whereas the analyzed values were 2.86, 4.80, 2.82, 1.40, 4.90, 1.56, 2.17, and 1.47 μg/kg.

### 2.6. Statistical Analysis

Because PAH data from this study did not fit into a normal distribution, the non-parametric Kruskal–Wallis tests or median tests for k independent samples were conducted to detect differences in data among the groups of samples.

Spearman correlation coefficients were used to analyze the correlations between the PAHs. A difference was regarded as significant when the value of P was below 0.05 and highly significant when it was less than 0.01.

The data were processed by a one-way analysis of variance with comparison of means (Tukey test) and with 95% confidence using Statistica 5.5 software (StatSoft Inc.,Tulsa, OK, USA). The software used was Microsoft Office Excel 2007 and SPSS 18.0 which can process and calculate the food sampling information, food consumption data, and concentration analysis results. With regard to PAH sample data that were lower than the limit of detection (LOD), half of the LOD was counted in the statistics.

### 2.7. Consumption Data

The daily ingestion amount (IR) (Table 2) and body weight (BW) (Table 3) were acquired from the data of the Chinese National Health and Nutrition Survey conducted in 2002.

### 2.8. Exposure Assessment

The PAH levels of 139 vegetable oils in China were collected for a risk assessment of PAH15. The risk assessment of PAHs was carried out only on the vegetable oils.

BaP was used as the representative PAH for the toxicity equivalent factors (TEFs) to express the relative carcinogenic risk of PAH15. The TEF was set to 1 for BaP and DBahA; 0.1 for BaA, BbF, BkF, and IcdP; 0.01 for Chr, BghiP, and Ant; and 0.001 for Pyr, Flt, Phe, Flo, Ace, and Nap. The BaPeq of vegetable oils was calculated according to Equation (1):(1)$$\mathrm{TEQBaP}=\sum_{i=1}^{n}\mathrm{Ci}\;\times \;𝑇𝐸𝐹𝑖$$
where Ci denotes the concentration of the PAH detected in the sample, and the TEFi denotes the TEF of congener (i) in the oil’s product. We used the sum of all TEQBaP (BaPeq)15 PAHs to assess the carcinogenic risk.

The chronic daily intake (CDI) of PAHs was calculated using the BaP equivalent concentration according to Equation (2) [32,33]:(2)$$\mathrm{CDI}\;(\mathrm{ng}\;\mathrm{BaPeq}/\mathrm{kg}\;\mathrm{bw}\;d)=\sum \frac{Ci\times IRi\times ED\times EF}{\mathrm{BW}\times \mathrm{AT}}$$
where Ci is the PAH15 calculated BaP equivalent concentration (ng BaPeq/g) for each vegetable oil; IRi is the daily intake of rapeseed oil, soybean oil, blended oil, olive oil, peanut oil, maize oil, and another oil for each individual’s sex and age (g/day). Exposure estimations were performed in four groups according to age: preschoolers (2–6 years old), school-age youths (7–12 years old), teenagers (13–17 years old), and adults (>18 years old). The ED is the exposure duration in years (preschoolers: ED = 5, school-age youths: ED = 6, teenagers: ED = 5, adults: ED = 53). The EF is the frequency exposure (365 days/years). The AT is the average exposure time (equal to 70 years for carcinogens, 25,550 days). IR and BW are approximately lognormal distribution (Table 2 and Table 3) [34]. Two PAHs (PAH2) are defined as the sum of BaP and Chr. Four PAHs (PAH4) are defined as the sum of BaP, Chr, BaA, and BbF. Eight PAHs (PAH8) are defined as the sum of BaP, Chr, BaA, BbF, BkF, DBahA, BghiP, and IcdP.

### 2.9. Risk Characterization

The MOE and ILCR were all evaluated. The Joint FAO/WHO Expert Committee on Food Additives (JECFA) and the European Food Safety Agency (EFSA) recommended using the MOE as an acceptable risk assessment method. To estimate the MOE, the ratio between the benchmark dose lower confidence limit (BMDL) and CDI was applied according to Equation (3) [33].
(3)$$\mathrm{MOE}=\frac{{\mathrm{BMDL}}_{10}}{\mathrm{CDI}}$$

Consistent with the method of the EFSA Scientific Committee, the CONTAM Panel used a range of statistical models to calculate the BMDL_10_ values for BaP, PAH2, PAH4, and PAH8. They chose the lowest BMDL_10_ values as 0.07, 0.17, 0.34, and 0.49 mg/kg bw/day, respectively [16]. In conclusion, MOEs ≥ 10,000 are classified as a low concern to human health. The ILCRs of PAHs in this study were calculated according to the following formula, Equation (4) [33]:(4)ILCR=CDI×SF×CF
where ILCR denotes the incremental lifetime cancer risk (dimensionless), SF denotes the oral cancer slope factor of BaP (geometric mean, 7.3 mg/kg/day), and CF denotes the conversion factor (10^−6^ mg/ng).

In our research, MOE and ILCR assessments due to the consumption of PAH-contaminated vegetable oils were calculated using MCS. The simulation was carried out by the mentioned parameters. The MCS model ran for 5000 iterations. Lastly, the means and P95(95th percentiles) of the MOE and ILCR distributions were selected to judge whether the exposed population was at risk.

## 3. Results and Discussion

Table 4 provides an overview of the incidence, means, media, P90, minimum, and maximum levels of PAH15 (BaA, Chr, BbF, BaP, BkF, DBahA, BghiP, IcdP, Pyr, Flt, Ant, Phe, Flo, Ace, and Nap, including PAH4, PAH8, and PAH15) in the vegetable oil and frying oil samples.

### 3.1. Levels of PAHs in Vegetable Oils

For the descriptive statistics, the PAHs’ concentrations below the LOD were calculated as a value of 0.15 μg/kg. As seen in Table 4, Nap showed the highest mean concentration and detection rates (61.08 μg/kg, 87.2%) in the non-carcinogenic PAHs, and comprised 35.6% of the total mean PAH concentration. However, Ant showed the lowest mean concentration and detection rates (4.62 μg/kg, 59.5%), and accounted for 2.7% of the total mean PAH concentration. Among the genotoxic PAHs (PAH8), Chr (3.37 μg/kg, 60.8%) and BaA (2.63 μg/kg, 46.6%) showed the highest mean values, followed by BbF (2.34 μg/kg, 53.4%) and BaP (2.16 μg/kg, 52.0%).

There were very large variations observed between the analytical samples with the concentration of BaP ranging from ND (not detected) to 25.5 μg/kg, the concentration of PAH4 ranging from ND to 123.5 μg/kg, the concentration of PAH8 ranging from 1.2 to 134.75 μg/kg, and the concentration of PAH15 ranging from 2.25 to 1670.9 μg/kg. The mean concentrations in samples were 3.37 μg/kg for Chr, 2.63 μg/kg for BaA, 2.34 μg/kg for BbF, 2.16 μg/kg for BaP, 10.49 μg/kg for PAH4, 14.63 μg/kg for PAH8, and 171.81 μg/kg for PAH15. The substance displaying the highest concentration was Chr. From 139 samples, 34 contained levels of PAH4 exceeding the EU limit of 10 μg/kg (24.5%). In addition, nine samples (6.5%) exceeded the GB-2762-2012 limit for BaP (10 μg/kg), thirty-three samples (23.7%) exceeded the EU limit for BaP (2 μg/kg), and twenty-two samples (15.8%) exceeded the EU limit for BaP (2 μg/kg) and PAH4 (10 μg/kg). The simultaneous over-limit ratio was 32.4% (45/139) (as long as more than 2 μg/kg of BaP or 10 μg/kg of PAH4 were considered to exceed the standard sample).

In terms of the type of sample, of the five types of vegetable oils (*n* > 20), the highest BaP contamination was observed in peanut oil (3.31 μg/kg), followed by rapeseed oil (2.40 μg/kg), maize oil (2.21 μg/kg), blend oil (1.94 μg/kg), and soybean oil (1.55 μg/kg) which were contaminated at low mean levels (Table 5). The mean levels of PAH4, PAH8, and PAH15 in peanut oil were also the highest in the common oils. In addition, seven olive oil samples were contaminated with PAHs at low mean levels (mean, 0.25 μg/kg), with only two samples contaminated with BaP at 1.1 μg/kg and 0.9 μg/kg. The relative contributions of each of the four PAHs to the total content of four PAHs in seven vegetable oils and frying oil are shown in Figure 1. Olive oil had the lowest BaP average contributions in vegetable oils. Maybe it has a more rigorous production process than other vegetable oils because of its high price.

The sample with the highest contamination levels of PAHs was a self-produced tea oil containing 22.1 μg/kg of BaP and 123.5 μg/kg of PAH4, which were significantly above the EU limits. In our study, even the mean concentrations of BaP (2.16 μg/kg) and PAH4 (10.49 μg/kg) were marginally above the EU maximum permitted levels in oils. Such a high PAH content suggests that manufacturers using conventional methods will make greater efforts to improve processing practices to decrease the possibility of PAH contamination in vegetable oils.

### 3.2. Levels of PAHs in Frying Oils (Vegetable Oils Used for Frying)

The increased PAH concentrations in frying oils compared to vegetable oils indicated the effect of frying. In the 48 frying oils, which were collected from restaurants, the mean contaminated levels of Chr, BaA, BbF, BaP, PAH4, PAH8, and PAH15 were 4.3 μg/kg, 3.92 μg/kg, 2.9 μg/kg, 3.59 μg/kg, 14.72 μg/kg, 19.74 μg/kg, and 310.1 μg/kg, respectively. The most contaminated sample was a fried oil containing 61 μg/kg of BaP, which showed values of 83.0 μg/kg for PAH4, 87.1 μg/kg for PAH8, and 3285.1 μg/kg for PAH15. After frying, the mean levels of Chr, BaA, BbF, and BaP in the oils increased significantly. For example, the BaP mean concentration in frying oils increased by 66% compared to oil purchased from the market (Table 4). It is important to point out that frying oils are usually a blend of oils or bean oil, and they are used because of their cheaper price. In addition, the mean concentrations of BaP in blend and bean oils were 1.94 μg/kg and 1.55 μg/kg, respectively, which were much lower than the frying oils. Regardless, frying increased the level of Bap, and the highest concentrations of PAHs were found after frying compared to grilling, roasting, and boiling.

Many studies have demonstrated the presence of PAHs in vegetable oils. However, there are few studies providing the levels of PAHs in frying oils used in restaurants in China. Other studies tested frying oils from fast-food restaurants set up in the laboratory, but not from restaurants in China [24,35].

Other investigators reported higher levels of the BaP and PAH4 in vegetable oils and oilseeds that ranged from 0.2 to 24.7 μg/kg and 1.1 to 112.7 μg/kg, respectively, with mean levels of 4.4 μg/kg and 19.3 μg/kg [36]. In this study, similar findings have been reported by investigators from other countries. According to a previous study, PAH4 concentrations in 36 samples of different brands ranged from 2.80 to 52.08 μg/kg, with 58% of the oils having PAH4 levels above the maximum value recommended by the European Commission [23]. Youssef and colleagues reported that 30% (12/40) of oil samples exceeded a concentration equal to the maximum residue limits (2 μg/kg for BaP), with an average BaP level of 1.95 μg/kg in Iran, consistent with the findings in China [37]. These results are also in agreement with Zhang and colleagues, who reported that the mean values of BaP and PAH4 were 2.32 μg/kg and 8.21 μg/kg, respectively [38].

However, the levels of PAHs in this study were substantially higher than those reported in previous studies [39,40]. For example, 0.41 and 0.51 μg/kg of BaP, as well as 1.35 and 2.05 μg/kg of PAH4, were detected in sesame and olive oils [41], and 3.63, 3.85, and 11.8 μg/kg of PAH4 were detected in soybean, olive, and rapeseed oils by Drabova and colleagues; Jiang and colleagues also reported a lower level of BaP in oils from Shandong, China, in which the mean level was 1.28 μg/kg [40,42]. These differences can be explained by different sampling strategies, geographic locations, or production processes. For example, different regions will affect the content of pollutants in the soil, and thus affect the concentration of pollutants in vegetable oil [43,44,45].

The levels of PAHs in vegetable oils vary greatly in oil types [23]. The average concentrations of PAH15 in different types of vegetable oils were different (Table 5). The concentrations of PAHs in plant roots were higher than those in plant leaves, proving that root uptake in soil is the main transfer route of PAHs from the environment to food [42,43,46]. For example, peanut oil was the most contaminated oil type, because peanut seeds grow in soil, unlike soybean and corn seeds which are leafy vegetables. In terms of the type of sample, peanut and rapeseed oil samples had higher levels of BaP, which are similar to our previous studies [26,38,42]. Moreover, PAHs in vegetable oils can be affected by refining and extraction procedures [47,48].

Moreover, it is noteworthy that, as a kind of cooking oil, there were two self-prepared tea oils with high contamination levels, with levels of 3.32 μg/kg and 22.1 μg/kg for BaP and 82.66 μg/kg and 123.5 μg/kg for PAH4. Concentrations of 6.79 μg/kg for BaP, 8.63 μg/kg for PAH4, and 21.23 μg/kg for PAH8 were found in rice oil.

In food processing, PAHs can be generated during heat treatment. Many studies have demonstrated that the high levels of PAHs were the result of frying at high temperatures [34,49]. It has been reported that the mean concentration of BaP from frying (105.9 μg/kg) was the highest, and the concentration of PAH4 was about 9.5 to 16.4 times the EU limit. Red pepper seed oils can also be contaminated by PAHs due to frying [50,51]. This conclusion is also supported by our data that the levels of PAHs were significantly increased in vegetable oils after deep-frying by restaurants. The type of fresh oil and the content of fat in food may affect the levels of PAHs in frying oils [40,52]. Moreover, the frying time is also an important factor [24,25]. Although these studies focused on the frying of fatty foods, no appreciable differences in the levels of PAHs were observed before and after the frying of non-fat foods [14].

### 3.3. Estimated Daily Intake

According to the TEQ values, the CDIs of PAHs due to the consumption of vegetable oils for each population group were calculated by Equations (1) and (2), as shown in Table 6.

The Mann–Whitney method was applied to detect differences between males and females of the same age, and no significant gender difference was detected (*p* > 0.05) in the four age groups, which indicates that differences in body weight and intake are not factors affecting the CID of PAHs. However, the results of the Kruskal–Wallis test revealed significant differences (*p* < 0.05) between different age groups. In general, across all age groups, the values for adults (18 years of age and older) were 5–10 times higher than the values of the other age groups for both males and females. Except for adults, we found that preschoolers (2–6 years of age) had slightly greater mean dietary exposure levels than school-age youths and teenagers.

Many previous studies compared the health risks of PAHs in vegetable oils. In this study, the mean dietary exposure level was estimated as previously reported by Yousefi and colleagues, where the mean (range) of CDI (PAH13) for adults who consumed corn, sunflower, blended oils, and frying was calculated as 0.49–0.54 (0.14–0.80) ng BaPeq/kg bw day, and that for children was calculated as 2.53–2.81 (0.73–4.13) ng BaPeq/kg bw day [23]. In Brazil, the consumption of soybean oils revealed the exposure level to PAH4 to be 7.3 ng BaPeq/kg bw day [53]. However, the average exposure levels of PAHs in this study were substantially higher than those of other studies. In Korea, the average exposure levels of BaPeq (PAH4) were 1.08 × 10^−2^, 1.13 × 10^−2^, 1.03 × 10^−2^, 4.58 × 10^−3^, and 1.59 × 10^−2^ ng BaPeq/kg bw day for sesame, perilla, red pepper seed, olive, and red pepper seasoning oils [41]. Zhang and colleagues reported that the median dietary exposure level of PAH4 using BaPeq in rapeseed oils ranged from 0.0580 to 0.3350 ng BaPeq/kg bw day for each age group in China [38].

Based on these data, the differences in dietary exposure to PAHs from different studies can be explained by differences in PAH concentrations and vegetable oil types. The various age groups also affected the results.

### 3.4. Risk Assessment

From the dietary exposure results on the analyzed vegetable oils, the risks of exposure to BaP, PAH2, PAH4, and PAH8 were characterized by calculating the MOEs, and that of PAH15 was characterized by calculating the ILCR for China.

The MOEs of the age/gender groups were performed through the mean and P95 estimates of dietary exposure to BaP, PAH2, PAH4, and PAH8 by Equation (3) (Table 7).

The EFSA recognized that the MOE values were greater than 1.0 × 10^4^, and they were considered a low public health concern. For the four age groups, adults had the lowest risk of exposure to PAHs, followed by preschoolers (2–6 years of age), school-age youths (7–12 years of age), and teenagers (13–17 years of age). Preschoolers had much lower body weight than school-age youths and teenagers, who had lower MOE values. The MOE values calculated using the 95th percentile dietary risk lowest-case MOE estimates reported in our study were above 1.0 × 10^6^ for PAH4 and PAH8 in adults, which were 104,649 and 115,378 for males and 103,737 and 114,122 for females, respectively. Indeed, these MOEs indicate low concerns for PAHs derived from vegetable oils.

The risk characterization of PAHs in vegetable oils by calculating MOE values has been reported for the general population in many countries. For example, Kang and colleagues reported the MOEs of PAHs from vegetable oils in Korea were 4,000,000 for BaP and 137,000 for PAH4, and thus concluded that the concern was negligible [54].

Yousefi and colleagues reported similar results that MOE values of vegetable oils in both children and adults were in a safe range (MOE ≥ 10,000) in Iran [23]. These findings summarized that PAHs did not pose a health risk for the population through the consumption of vegetable oils, because MOE values were greater than 10,000.

The US EPA (United States of Environmental Protection Agency) recognized that ILCR values lower than 1 × 10^−6^ were negligible, and ILCR values higher than 1 × 10^−4^ were likely harmful to people’s health. Furthermore, ILCR values indicated a tolerable risk within a range of 1 × 10^−6^ to 1 × 10^−4^ [33]. The cumulative probability distributions of the ILCR values for population groups in China are showed in Figure 2.

For males (A), the mean ILCR values were estimated to be 3.27 × 10^−6^, 3.07 × 10^−6^, 1.73 × 10^−6^, and 1.53 × 10^−5^ for preschoolers, school-age youths, teenagers, and adults, respectively; for females (B), the mean ILCR values were estimated to be 3.27 × 10^−6^, 2.91 × 10^−6^, 1.55 × 10^−6^, and 1.55 × 10^−5^, respectively. The cumulative probabilities of the ILCRs of different age groups were less than the priority risk level (10^−4^). Furthermore, 2.5% to 36.5% of ILCRs were below the lower limit of the safe acceptable range (10^−6^) in the four age groups of males and females. In terms of gender, there was no difference in the ILCR values in all age groups between males and females. However, the carcinogenic risk differed among the age groups. The ranking of ILCR values based on age in decreasing order was as follows: adults, preschoolers, school-age youths, and teenagers for both males and females. The adult age group was exposed to the highest carcinogenic risk. For adult males, 97.3% of the cumulative probabilities of ILCRs fell within the range of 10^−6^ to 10^−4^, which indicated a slight potential carcinogenic risk, consistent with Li et al. and Zhang et al. and Jiang et al. [34,38,42]. In addition, Gelavizh found that the consumption of edible oils was safe through the risk assessment by MOE and ILCR in Iran [55]. Adults were exposed to the greatest cancer risk, which could be attributed to the longest exposure duration and the highest consumption of vegetable oil. Although preschoolers have lower dietary intake than school-age children and adolescents, preschoolers have much lower body weight, resulting in higher ILCR values. Preschoolers were sensitive to the health risks of pollutants, and thus, more attention should be given to their health issues, in agreement with the results of a previous study which showed that children are more vulnerable to PAH exposure and have greater risks of developing health issues [21].

## 4. Conclusions

This study investigated the profiles and levels of PAH15 in vegetable oils and frying oils (after repeated frying by restaurants). The data indicated that vegetable oils are highly contaminated, with 32.4% of vegetable oils from China exceeding the EU standard limit of the levels of PAHs. Only 6.5% of the samples surpassed the GB-2762-2012 limit for BaP (10 μg/kg). Based on the results, the following mean levels of BaP in the mainly vegetable oils in China were established: soybean oil < blend oil < rapeseed oil < peanut oil. The levels of PAH15 detected in two produced tea oils were the highest. The PAH15 levels were significantly higher in frying oils from restaurants compared to vegetable oils. The MOE and ILCR values for PAH15 for four different age groups corresponded to low risk levels in China. However, 2–6 year old children were more vulnerable than older children and adolescents.

PAHs have also been shown to be present in fruits, cereals, vegetables, milks, and smoked and roasted food. Consumers are at considerable carcinogenic risk through milk and milk powders [55]. The exposure via these food items should be studied in the near future.

## Figures and Tables

**Figure 1 foods-12-00811-f001:**
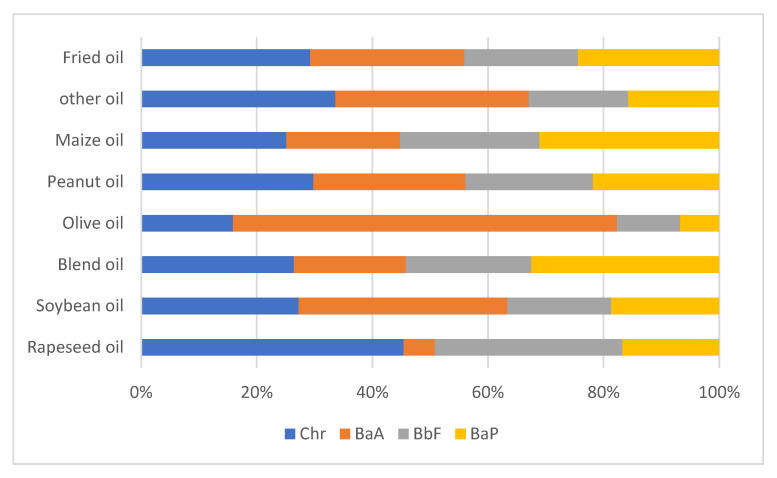
Relative contribution of 4 PAHs in 7 vegetable oils’ categories and fried oil.

**Figure 2 foods-12-00811-f002:**
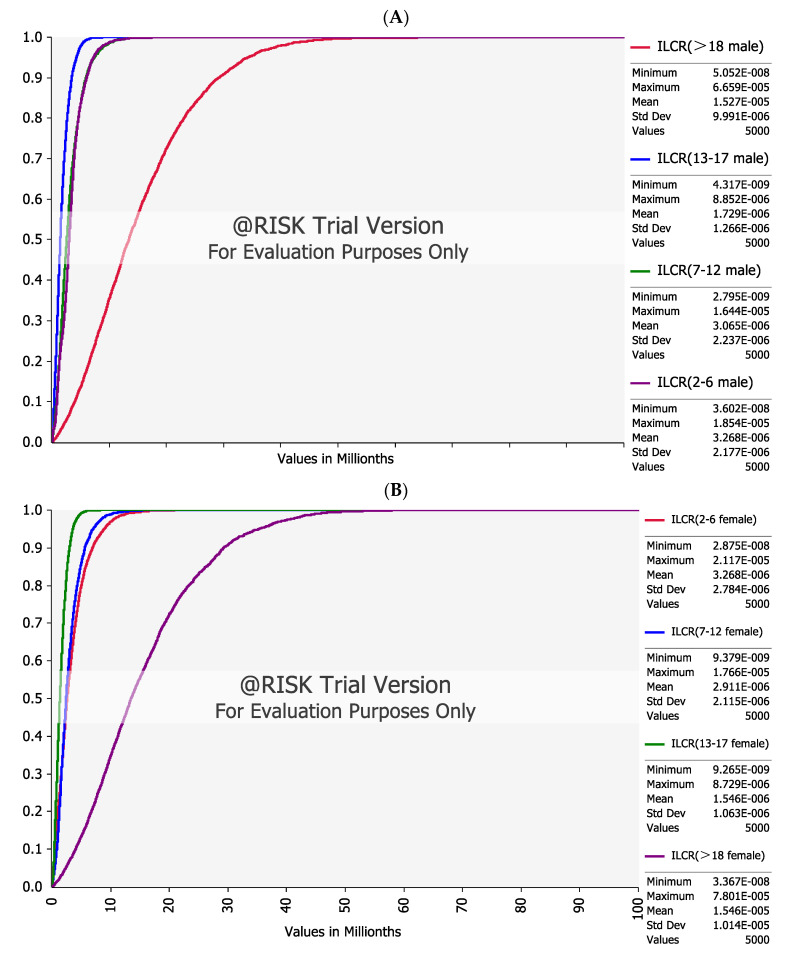
Cumulative probability of ILCR from vegetable oils in China. (**A**): male; (**B**): female.

**Table 1 foods-12-00811-t001:** Fluorescence detection settings for the determination of PAHs.

PAHs	Time Window (min)	λ Excitation (nm)	λ Emission (nm)
Nap, Ace, Flo	0	270	324
Phe, Ant	11.8	248	375
Flt	14.0	280	462
Pyr BaA Chr	14.9	270	446
BbF	19.0	256	446
BkF BaP DBahA BghiP	20.9	292	410
IcdP	24.7	290	355

**Table 2 foods-12-00811-t002:** Estimated lognormal probability density functions describing the ranges of possible oil consumption rates (g/day) for groups in the study.

Age	No.	Rapeseed Oil		Soybean Oil		Blend Oil		Olive Oil		Peanut Oil		Maize Oil		Other Oil		Total	
		Mean	P95	Mean	P95	Mean	P95	Mean	P95	Mean	P95	Mean	P95	Mean	P95	Mean	P95
2–6 (male)	75	19.15	57.83	2.94	31.80	0.00	0.00	0.00	0.00	2.70	25.20	1.81	14.46	4.99	32.28	31.60	65.64
2–6 (female)	65	14.12	59.91	3.29	29.20	0.11	0.00	0.00	0.00	6.72	40.76	1.83	4.00	4.43	28.14	30.49	75.88
7–12 (male)	479	14.81	60.22	3.19	29.27	0.12	0.00	0.00	0.00	5.04	38.16	0.29	0.00	9.22	43.12	32.66	71.57
7–12 (female)	472	14.89	60.95	3.83	33.82	0.11	0.00	0.00	0.00	3.70	31.45	1.04	0.00	8.64	39.78	32.22	70.05
13–17 (male)	224	16.75	64.75	4.48	35.85	0.00	0.00	0.00	0.00	3.34	28.92	0.61	0.00	11.74	59.17	36.92	81.76
13–17 (female)	219	16.70	61.77	3.40	29.26	0.00	0.00	0.00	0.00	2.57	24.49	1.40	0.00	8.91	44.33	32.99	69.01
>18 (male)	5760	14.24	66.80	4.48	36.83	0.27	0.00	0.22	0.00	6.80	47.51	1.59	0.00	10.46	56.87	38.06	82.86
>18 (female)	6607	13.95	66.62	4.46	36.33	0.28	0.00	0.21	0.00	6.91	47.58	1.64	0.00	10.54	56.44	38.00	82.50

**Table 3 foods-12-00811-t003:** Estimated lognormal probability density functions describing the ranges of body mass (kg) for groups in the study.

Gender	Preschoolers (2–6)	School-Agers (7–12)	Youths (13–17)	Adults (>18)
female	23.02 ± 6.23	32.52 ± 10.49	50.40 ± 13.21	60.88 ± 11.12
male	22.06 ± 4.35	31.52 ± 9.50	50.33 ± 12.15	61.64 ± 11.17

**Table 4 foods-12-00811-t004:** Occurrence and contamination levels of PAHs in vegetable and frying oils in China.

	Parameter	BaP	Chr	BaA	BbF	BkF	DBahA	BghiP	IcdP	Pyr	Flt	Ant	Phe	Fluo	Ace	Nap	PAH4	PAH8	PAH15
vegetable oil	Incidence > LOD(%)	52	60.8	46.6	53.4	41.9	34.5	45.3	22.3	83.1	70.9	59.5	92.6	76.4	50	87.2	/	/	/
	Mean	2.16	3.37	2.63	2.34	1.18	0.91	1.36	0.7	9.43	16.78	4.62	31.84	13.25	20.18	61.08	10.49	14.63	171.81
	Media	0.4	0.98	0.15	0.46	0.15	0.15	0.15	0.15	4.96	2.63	0.98	14.05	6.34	0.35	20.45	3.94	6.895	122.67
	P90	6.18	7.6	5.68	5.14	3.35	2.8	3.84	1.65	25.3	40.52	10.05	73.61	23.78	50.6	167.5	28.54	37.47	347.42
	MIN	0.15	0.15	0.15	0.15	0.15	0.15	0.15	0.15	0.15	0.15	0.15	0.15	0.15	0.15	0.15	0.6	1.2	2.25
	MAX	25.5	45.1	60.4	52.6	15.4	9.47	22.8	10.5	104	394	121	488	190	293	1072	123.5	134.75	1670.9
frying oil	Incidence > LOD(%)	64.6	70.8	64.6	52.1	58.3	50	47.9	39.6	68.8	85.4	75	93.8	89.6	68.8	91.7	/	/	/
	Mean	3.59	4.3	3.92	2.9	1.68	0.86	1.6	0.88	62.32	21.77	7.18	42.52	22.37	11.74	122.45	14.72	19.74	310.1
	Media	0.5	1.2	0.8	0.61	0.48	0.22	0.5	0.25	1.82	3.19	3.2	18.3	9.2	3.5	14	5.01	8.43	121.74
	P90	8.37	14.26	7.96	6.18	4.35	1.91	3.68	2.13	33.43	34.85	17.34	66.38	43.78	17.86	443.7	33.39	45.08	652.71
	MIN	0.05	0.1	0.1	0.1	0.05	0.1	0.1	0.1	0.15	0.15	0.1	0.25	0.05	0.15	0.15	0.4	0.7	3.75
	MAX	61	42.2	74.4	32.1	12.2	6.89	11.4	6	1299	336	79.6	565	267	121	716	143	152.21	3285.06

Genotoxic 2 PAHs include the sum of benzo[*a*]pyrene and chrysene. Genotoxic 4 PAHs include the sum of benzo[*a*]pyrene, chrysene, benz[*a*]anthracene, and benzo[*b*]fluoranthene. Genotoxic 8 PAHs include the sum of benz[a]anthracene, chrysene, benzo[*b*]fluoranthene, benzo[*k*]fluoranthene, benzo[*a*]pyrene, indeno[1,2,3-cd]pyrene, dibenz[*a*,*h*]anthracene, and benzo[*g,h,i*]perylene.

**Table 5 foods-12-00811-t005:** PAH concentrations by BaP and sum of PAH4, PAH8, PAH15, and TEQ_Bap_ as BaP concentration estimated by TEFs in different types of vegetable and frying oils.

Oil Tyle	No. Samples	BaP		PAH4		PAH8		PAH15		TEQ_Bap_	
		Mean	Range	Mean	Range	Mean	Range	Mean	Range	Mean	Range
Rapeseed oil	20	2.40	<RL-25.5	14.32	<RL-118.85	17.98	<RL-134.7	141.54	<RL-607.45	4.01	0.37–33.59
Soybean oil	28	1.55	<RL-10.3	8.28	<RL-47.46	11.97	<RL-66.11	103.82	7.99–360.42	3.38	0.37–21.2
Blend oil	23	1.94	<RL-21.1	5.98	<RL-30.2	8.80	<RL-38.4	213.82	<RL-1314.05	3.41	0.37–22.85
Olive oil	7	0.39	<RL-1.1	5.83	<RL-32.37	9.16	<RL-34.91	154.49	24.67–287.35	2.01	00.47–4.18
Peanut oil	21	3.31	<RL-17.1	15.18	<RL-88.32	20.41	<RL-103.36	278.72	24.96–1022.01	5.43	0.39–20.63
Maize oil	20	2.21	<RL-11.9	7.12	<RL-27.88	11.31	1.44–40.02	143.00	9.95–762.25	4.11	0.38–14.46
Other oil	20	2.36	<RL-22.1	15.01	1.44–123.5	20.91	2.41–134.75	171.59	5.88–1670.9	4.58	0.38–32.05
Frying oil	48	3.37	<RL-24.4	10.9	0.3–129.50	18.30	0.70–152.21	246.81	3.75–2160.21	4.53	0.37–29.64

**Table 6 foods-12-00811-t006:** CDIs of PAHs (ng BaPeq/kg bw day) for average body masses using the observed mean concentration of BaP, PAH2, PAH4, PAH8, and PAH15 for each species based on different daily consumption of vegetable oils by males and females.

Age	BaP		PAH2		PAH4		PAH8		PAH15	
	Mean	P95	Mean	P95	Mean	P95	Mean	P95	Mean	P95
2–6 (male)	0.244	0.539	0.249	0.549	0.307	0.669	0.407	0.867	0.426	0.907
2–6 (female)	0.236	0.631	0.240	0.647	0.296	0.795	0.391	1.041	0.411	1.090
7–12 (male)	0.217	0.496	0.222	0.503	0.277	0.625	0.366	0.816	0.384	0.854
7–12 (female)	0.202	0.443	0.207	0.454	0.361	0.787	0.345	0.751	0.361	0.787
13–17 (male)	0.124	0.295	0.127	0.301	0.159	0.385	0.212	0.509	0.222	0.540
13–17 (female)	0.110	0.249	0.113	0.256	0.141	0.308	0.188	0.401	0.197	0.425
>18 (male)	1.135	2.544	1.159	2.603	1.445	3.249	1.931	4.247	2.026	4.466
>18 (female)	1.149	2.575	1.172	2.630	1.463	3.278	1.954	4.294	2.051	4.505

**Table 7 foods-12-00811-t007:** MOEs for age/gender groups were calculated by dividing the lowest BMDL_10_ values among the models with acceptable fits by the mean and 95th percentile estimates of dietary exposure to BaP, PAH2, PAH4, and PAH8.

Age	BaP		PAH2		PAH4		PAH8	
	Mean	P95	Mean	P95	Mean	P95	Mean	P95
2–6 (male)	287,331	129,796	682,822	309,666	1,106,062	507,993	1,202,838	565,324
2–6 (female)	297,110	110,945	707,412	262,873	1,147,686	427,482	1,251,968	470,923
7–12 (male)	322,775	141,188	767,165	338,271	1,228,542	544,341	1,337,984	600,647
7–12 (female)	345,760	158,029	821,743	374,764	1,317,361	600,400	1,422,205	652,464
13–17 (male)	565,447	237,560	1,343,280	564,872	2,142,262	882,870	2,308,208	962,884
13–17 (female)	633,619	280,790	1,504,933	665,110	2,413,834	1,105,143	2,604,790	1,220,536
>18 (male)	61,658	27,516	146,729	65,305	235,263	104,649	253,796	115,378
>18 (female)	60,922	27,190	144,991	64,646	232,470	103,737	250,733	114,122

## Data Availability

Data are available from the corresponding author upon request.

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
