# Peer review of "Levels and Health Risk Assessment of Polycyclic Aromatic Hydrocarbons in Vegetable Oils and Frying Oils by Using the Margin of Exposure (MOE) and the Incremental Lifetime Cancer Risk (ILCR) Approach in China"

_foods, 2023, doi:10.3390/foods12040811_

Round 1

Reviewer 1 Report

See attached file.

Author Response

Response to Reviewer 1 Comments

Lines 2–5: Rewrite to “Health risk assessment using the margin of exposure (MOE) and incremental lifetime cancer risk (ILCR) of the polycyclic aromatic hydrocarbon levels in vegetable oils and frying oils in China”

Response: It has been rewritten. “Health risk assessment of the polycyclic aromatic hydrocarbons levels in vegetable oils and frying oils in China

Line 11: Change “139 edible oils” to “139 vegetable oils” Note: please be consistent with terminology.

Response: It has been changed.

Question for the Authors: Are frying oils not edible?

Response: They are the vegetable oils after frying food. For example, soybean oil used in fried chicken in restaurants. Then we analyzed the used soybean oil. Frying oils are edible, but in family life people don’t usually eat them.

Line 26–27: Change “They are widely existed in” to “They widely exist in”

Response: It has been changed.

Line 34: Change “the increase levels” to “the increased levels”

Response: It has been changed.

Line 38: delete one period.

Response: It has been deleted.

Line 39: Change “possess as carcinogenic,” to “possess carcinogenic,”

Response: It has been changed.

Line 41: Change “that PAHs exposure can” to “that exposure to PAHs can”

Response: It has been modified.

Line 46: Change “that PAHs . . . asthma [20].” To “that childhood asthma is associated with exposure to PAHs [20].”

Response: It has been changed.

Line 50: Define BaP, Chr, BaA and BbF at first mention of the acronyms.

Response: It has been modified.

Lines 50–51: Change “, whereas China” to “; whereas, China”

Response: It has been changed.

Line 54: Change “oils in PAHs” to “oils containing PAHs”

Response: It has been changed.

Line 58: Change “and the intake” to “and intake”

Response: It has been changed.

Line 64: Change “elevated PAHs lev”

to “elevated lev”

Response: It has been changed.

Line 65: Change “els are” to “els of PAHs are”

Response: It has been changed.

Lines 65–66: Change “It should be a useful work to” to “It would be useful to”

Response: It has been changed.

Line 66: Change “oils which are in the restaurants” to “oils used in restaurants”

Response: It has been changed.

Line 79: Change “including edible oils” to “including vegetable oils”

Response: It has been changed.

Lines 83–84: At the first mention of acronyms, they should be defined. “BkF — Nap”

Response: It has been modified.

Line 94: Remove the space: Change “acetate: cyclohexane” to “acetate:cyclohexane”

Line 94: Change “sample was” to “samples were”

Response: It has been changed.

Line 99: Change “acetate: cyclohexane: (1:1, v/v)” to “acetate:cyclohexane (1:1, v/v)”

Response: It has been changed.

Line 101: include the name of the vacuum rotary evaporation instrument, manufacturer name, and city, state, and country.

Response: It has been added. “a vacuum rotary evaporator (Rotavapor R-300, Buchi, Switzerland)

Line 103: Change “centrator, re-dissolved” to “centrator, and re-dissolved”

Response: It has been changed.

Line 104: Change “for HPLC-FLD.” To “for determination by HPLC-FLD.”

Response: It has been changed.

Line 108: Change “on fluorescence” to “using fluorescence”

Response: It has been changed.

Line 116: Change “between standard” to “between the standard”

Response: It has been modified.

Line 116: Change “on a given” to “on any given”

Response: It has been changed.

Line 118: Change “the detected procedure” to “the detection procedure”

Response: It has been changed.

Line 120: Change “the comparability” to “comparability”

Response: It has been changed.

Line 120: Change “data from different” to “data between different”

Response: It has been modified.

Line 124: Change “blanked edible oil” to “blank vegetable oil”

Response: It has been changed.

Line 126: Change “all analysts.” To “all analyses.”

Response: It has been changed.

Line 134: Change “into the normal” to “into a normal”

Response: It has been changed.

Line 134: Change “, non-par” to “, the non-par”

Response: It has been changed.

Line 144: Change “that are lower” to “that were lower”

Response: It has been changed.

Line 155: Change “139 edible oils” to “139 vegetable oils”

Response: It has been changed.

Line 156: “There is actually no frying oil consumption.” Note: I do not believe this casual statement. If these oils are used to cook (fry) foods, then the foods will contain substantial amounts of these frying oils.

Authors, please remove this sentence or change it appropriately.

Response: It has been removed.

It is difficult to obtain the consumption data on frying oil in actually. There must be frying oil consumption, but there is no consumption data. (The consumption data can be used in the risk assessment.)

Line 158: Change “The representative of PAHs is BaP, which was used as toxicity” to “BaP was used as the representative PAH for the toxicity”

Response: It has been changed.

Line 167: Change “using BaP” to “using the BaP”

Response: It has been changed.

Line 172: Change “is PAH15 calculated as BaP” to “is the PAH15 calculated BaP”

Response: It has been changed.

Line 183: Change “DBahA,     BghiP” to “DBahA, BghiP”

Response: It has been changed.

Line 187: Change “MOE as an acceptable” to “using the MOE as an acceptable”

Response: It has been changed.

Line 194: Change “Panel using a” to “Panel used a”

Response: It has been changed.

Line 194: Change “models calculated” to “models to calculate”

Response: It has been changed.

Line 196: Change “are clarified as low” to “are classified as a low”

Response: It has been changed.

Line 214: Change “PAHs concentrations” to “the PAH concentrations”

Response: It has been changed.

Line 215: Change “0.15 mg/ kg.” to “0.15 mg/kg.”

Response: It has been changed.

Line 223: Change “benzo[a]pyrene” to “benzo[a]pyrene”

Response: It has been changed.

Line 223 and 224: Add spaces after the periods.

Response: It has been changed.

Line 224: Italicize the letters: “[a]pyrene, [a]anthracene, and benzo[b]” to “[a]pyrene, [a]anthracene, and benzo[b]”

Response: It has been changed.

Line 225: Italicize the letters: “benz[a], benzo[b], benzo[k]” to “benz[a], benzo[b], benzo[k]”

Response: It has been changed.

Line 226: Italicize the letters: “benzo[a], [1,2,3-cd]pyrene, dibenz[a,h], and benzo[ghi]perylene” to “benzo[a], [1,2,3-cd]pyrene, dibenz[a,h], and benzo[ghi]perylene”

Response: It has been changed.

Line 227: Change “analyted” to “analytical”

Response: It has been changed.

Line 233: Remove the extra spaces: “Chr.     From” to “Chr. From”

Response: It has been changed.

Line 260: Change “PAHs” to “PAH”

Response: It has been changed.

Line 262: Change “oils be used” to “oils used”

Response: It has been changed.

Line 271: Change “important point” to “important to point”

Response: It has been changed.

Line 271: Change “are usually” to “are usually a”

Response: It has been changed.

Line 272: Change “blend oil or” to “blend of oils or”

Response: It has been changed.

Line 278: Change “studies on the” to “studies providing the”

Response: It has been changed.

Line 286: Change “samples in a concentration exceeded the maximum” to “samples exceeded a concentration equal to the maximum”

Response: It has been changed.

Line 287: Change “consistent with” to “consistent with the”

Response: It has been changed.

Line 300: Change “in the vegetable oil” to “in vegetable oils”

Response: It has been changed.

Line 308: Change “edible oils” to “vegetable oils”

Response: It has been changed.

Line 320: Change “edible oils” to “vegetable oils”

Response: It has been changed.

Line 337: Change “at average” to “for average”

Response: It has been changed.

Line 340: Change “in edible oils” to “in vegetable oils”

Response: It has been changed.

Line 356: Change “edible oils” to “vegetable oils”

Response: It has been changed.

Line 365: Change “considered as low concern from the public health.” To “considered a low public health concern.”

Response: It has been changed.

Line 366: Change “exposure the PAHs,” to “exposure to PAHs,”

Response: It has been changed.

Lines 375, 377, and 379: Change “edible oils” to “vegetable oils”

Response: It has been changed.

Line 411: Change “PAHs” to “PAH”

Response: It has been changed.

Line 428: Change “on that foods.” to “on those cooking methods.”

Response: It has been changed.

Line 436: Change “food-borne” to “foodborne”

Response: It has been changed.

Reviewer 2 Report

Comments to the Author

The methodology is good.

In my opinion, the English language expressions in the manuscript require copyediting. The writing of this manuscript is not easy for readers to understand. Some sentences fail to express the original meaning. I suggest using a professional copyeditor or a native English speaker to copy edit this manuscript. 

Abstract:

Abstract section should be improved.

In all text, edible oil should be changed to vegetable oil. Is frying oil non-edible?

Mention the full word of EPA, EU and BaP.

- Mention the compounds in PAH4 and PAH8

Mention the method information such as recovery, IOD and LOQ in this section.

Enter the full name for the first time and abbreviate it in brackets and then just enter the abbreviation in the text

- Sort by alphabetical order.

- It is better to use keywords other than the title

Introduction

Explain a little about vegetable oils and frying oils in this section.

- Explain a little about the ways of entering and producing these compounds (PAHs) in food.

- It is necessary to give explanations about the preparation method and equipment such as HPLC, GC, etc.

The logic of the current introduction should be revised, and I suggest organizing the Introduction section as following order: importance and meanings, previous studies (literature review), the gaps of previous studies, and objectives of this study.

The objectives were too general. Therefore, this part of the manuscript must be rewritten to be
more detailed.

Please see the following articles and to make the article more complete:

https://doi.org/10.1016/j.jfca.2019.103331

Materials and reagents

How much is the sample size exactly?

2017? After about 5 years, are you planning to publish now?

Where is LOD , LOQ ØŸHave you had only one LOD&LOQ?

Result and Discussion

The units in this section should be the same

It is necessary to compare with more studies and summarize the results of other studies.

Conclusion

Conclusions should be clear and concise

Please write the limitations of the study

Throughout the text, some words must match the format of the journal.

In general, the text has many errors that must be carefully corrected by the author.

Use newer references

Author Response

Response to Reviewer 2 Comments

Comments to the Author

The methodology is good.

In my opinion, the English language expressions in the manuscript require copyediting. The writing of this manuscript is not easy for readers to understand. Some sentences fail to express the original meaning. I suggest using a professional copyeditor or a native English speaker to copy edit this manuscript.

Abstract:

Abstract section should be improved.

In all text, edible oil should be changed to vegetable oil. Is frying oil non-edible?

Response: I have changed edible oil to vegetable oil.

Frying oil is the vegetable oils after frying food. For example, soybean oil used in fried chicken in restaurants. Then we analyzed the used soybean oil. Frying oils are edible, but in family life people don’t usually eat them.

Mention the full word of EPA, EU and BaP.

Response: I have mentioned the full word of EPA, EU and BaP. Environmental Protection Agency (EPA); European Union (EU); benzo[a]pyrene (Bap)

- Mention the compounds in PAH4 and PAH8

Response: I have modified the full word of the compounds in PAH4 and PAH8.

Mention the method information such as recovery, IOD and LOQ in this section.

Response: I have added the recovery, LOD and LOQ.

Enter the full name for the first time and abbreviate it in brackets and then just enter the abbreviation in the text

Response: I have entered the full name.

- Sort by alphabetical order.

Response: I have sorted by alphabetical order.

- It is better to use keywords other than the title

Response: I have modified the title and keywords.

Introduction

Explain a little about vegetable oils and frying oils in this section.

Response: I have explained.

- Explain a little about the ways of entering and producing these compounds (PAHs) in food.

Response: I have explain it. (PAHs in the atmosphere, soil and water can be absorbed by plants, and then transferred to animals through the food chain. Therefore, raw foods contain PAHs. Processing processes (such as baking, frying and smoking) at high temperatures are also major sources generating PAHs in food)

- It is necessary to give explanations about the preparation method and equipment such as HPLC, GC, etc.

Response: I have explanted it in the introduction. (Many methods was performed to detect and measure the PAHs in different food products, using GC–MS/FID (gas chromatography equipped with mass spectrometry or flame ionization detector), HPLC-UV/FID (high-performance liquid chromatography equipped with ultraviolet or fluorescence detectors). Extraction PAHs from food is crucial in the multi-stage sample preparation procedure. Gel permeation chromatography combines separation and purification, and the method is simple, rapid, easy to operate, automatic, high recovery rate, good repeatability and good stability. It is widely used in the separation and purification of PAHs in vegetable oils.

The logic of the current introduction should be revised, and I suggest organizing the Introduction section as following order: importance and meanings, previous studies (literature review), the gaps of previous studies, and objectives of this study.

Response: I have revised as suggested.

The objectives were too general. Therefore, this part of the manuscript must be rewritten to be

more detailed.

Response: I have revised.

Please see the following articles and to make the article more complete:

https://doi.org/10.1016/j.jfca.2019.103331

Response: I have saw this articles.

Materials and reagents

How much is the sample size exactly?

A total of 148 samples, including edible oils (139) and frying oils (48)

Response: I have revised. A total of 139 samples, including edible oils (91) and frying oils (48).

2017? After about 5 years, are you planning to publish now?

Response: The samples were collected previously in China's national risk monitoring plan. Now this dates were analyzed.

Where is LOD , LOQ ØŸHave you had only one LOD&LOQ?

Response: They are in the “Method validation”. The limit of detection and limit of quantitation were ranged between 0.2-0.3 and 0.6-1 μg/kg, respectively.

Result and Discussion

The units in this section should be the same

Response: I have revised and use the same units.

It is necessary to compare with more studies and summarize the results of other studies.

Response: I have added and compared some more new studies results.

Conclusion

Conclusions should be clear and concise

Response: I have revised.

Please write the limitations of the study

Response: I write the limitations of the study.

Throughout the text, some words must match the format of the journal.

Response: I have revised.

In general, the text has many errors that must be carefully corrected by the author.

Response: I have revised.

Use newer references

Response: I have used newer references.

Round 2

Reviewer 2 Report

Accept